# GLAD: Learning Sparse Graph Recovery

Harsh Shrivastava[1]        Xinshi Chen[2]        Binghong Chen[1]        Guanghui Lan[3]
Srinivas Aluru[1]        Han Liu[4]        Le Song[1,5]

{[1]School of Computational Science & Engineering, [2]School of Mathematics,
[3]School of Industrial and Systems Engineering }at Georgia Institute of Technology,
[4]Computer Science Department at Northwestern University, [5]Ant Financial Services Group

## Abstract

Recovering sparse conditional independence graphs from data is a fundamental problem in machine learning with wide applications. A popular formulation of the problem is an $\ell_1$ regularized maximum likelihood estimation. Many convex optimization algorithms have been designed to solve this formulation to recover the graph structure. Recently, there is a surge of interest to learn algorithms directly based on data, and in this case, learn to map empirical covariance to the sparse precision matrix. However, it is a challenging task in this case, since the symmetric positive definiteness (SPD) and sparsity of the matrix are not easy to enforce in learned algorithms, and a direct mapping from data to precision matrix may contain many parameters. We propose a deep learning architecture, GLAD, which uses an Alternating Minimization (AM) algorithm as our model inductive bias, and learns the model parameters via supervised learning. We show that GLAD learns a very compact and effective model for recovering sparse graphs from data.

## 1 Introduction

Recovering sparse conditional independence graphs from data is a fundamental problem in high dimensional statistics and time series analysis, and it has found applications in diverse areas. In computational biology, a sparse graph structure between gene expression data may be used to understand gene regulatory networks; in finance, a sparse graph structure between financial time-series may be used to understand the relationship between different financial assets. A popular formulation of the problem is an $\ell_1$ regularization log-determinant estimation of the precision matrix. Based on this convex formulation, many algorithms have been designed to solve this problem efficiently, and one can formally prove that under a list of conditions, the solution of the optimization problem is guaranteed to recover the graph structure with high probability.

However, convex optimization based approaches have their own limitations. The hyperparameters, such as the regularization parameters and learning rate, may depend on unknown constants, and need to be tuned carefully to achieve the recovery results. Furthermore, the formulation uses a single regularization parameter for all entries in the precision matrix, which may not be optimal. It is intuitive that one may obtain better recovery results by allowing the regularization parameters to vary across the entries in the precision matrix. However, such flexibility will lead to a quadratic increase in the number of hyperparameters, but it is hard for traditional approaches to search over a large number of hyperparameters. Thus, a new paradigm may be needed for designing more effective sparse recovery algorithms.

Recently, there has been a surge of interest in a new paradigm of algorithm design, where algorithms are augmented with learning modules trained directly with data, rather than prescribing every step of the algorithms. This is meaningful because very often a family of optimization problems needs to be solved again and again, similar in structures but different in data. A data-driven algorithm may be able to leverage this distribution of problem instances, and learn an algorithm which performs better than traditional convex formulation. In our case, the sparse graph recovery problem may also need to be solved again and again, where the underlying graphs are different but have similar degree distribution, the magnitude of the precision matrix entries, etc. For instance, gene regulatory networks may be rewiring depending on the time and conditions, and we want to estimate them from gene

expression data. Company relations may evolve over time, and we want to estimate their graph from stock data. Thus, we will also explore data-driven algorithm design in this paper.

Given a task (e.g. an optimization problem), an algorithm will solve it and provide a solution. Thus we can view an algorithm as a function mapping, where the input is the task-specific information (i.e. the sample covariance matrix in our case) and the output is the solution (i.e. the estimated precision matrix in our case). However, it is very challenging to design a data-driven algorithm for precision matrix estimation. First, the input and output of the problem may be large. A neural network parameterization of direct mapping from the input covariance matrix to the output precision matrix may require as many parameters as the square of the number of dimensions. Second, there are many structure constraints in the output. The resulting precision matrix needs to be positive definite and sparse, which is not easy to enforce by a simple deep learning architecture. Third, direct mapping may result in a model with lots of parameters, and hence may require lots of data to learn. Thus a data-driven algorithm needs to be designed carefully to achieve a better bias-variance trade-off and satisfy the output constraints.

In this paper, we propose a deep learning model 'GLAD' with following attributes:

- Uses an unrolled Alternating Minimization (AM) algorithm as an inductive bias.
- The regularization and the square penalty terms are parameterized as entry-wise functions of intermediate solutions, allowing GLAD to learn to perform entry-wise regularization update.
- Furthermore, this data-driven algorithm is trained with a collection of problem instances in a supervised fashion, by directly comparing the algorithm outputs to the ground truth graphs.

In our experiments, we show that the AM architecture provides very good inductive bias, allowing the model to learn very effective sparse graph recovery algorithm with a small amount of training data. In all cases, the learned algorithm can recover sparse graph structures with much fewer data points from a new problem, and it also works well in recovering gene regulatory networks based on realistic gene expression data generators.

**Related works.** Belilovsky et al. (2017) considers CNN based architecture that directly maps empirical covariance matrices to estimated graph structures. Previous works have parameterized optimization algorithms as recurrent neural networks or policies in reinforcement learning. For instance, Andrychowicz et al. (2016) considered directly parameterizing optimization algorithm as an RNN based framework for learning to learn. Li & Malik (2016) approach the problem of automating algorithm design from reinforcement learning perspective and represent any particular optimization algorithm as a policy. Khalil et al. (2017) learn combinatorial optimzation over graph via deep Q-learning. These works did not consider the structures of our sparse graph recovery problem. Another interesting line of approach is to develop deep neural networks based on unfolding an iterative algorithm Gregor & LeCun (2010); Chen et al. (2018); Liu et al. (2018). Liu et al. (2018) developed ALISTA which is based on unrolling the Iterative Shrinkage Thresholding Algorithm (ISTA). Sun et al. (2016) developed 'ADMM-Net', which is also developed for compressive sensing of MRI data. Though these seminal works were primarily developed for compressive sensing applications, they alluded to the general theme of using unrolled algorithms as inductive biases. We thus identify a suitable unrolled algorithm and leverage its inductive bias to solve the sparse graph recovery problem.

## 2 Sparse Graph Recovery Problem and Convex Formulation

Given $m$ observations of a $d$-dimensional multivariate Gaussian random variable $X = [X_1, \ldots, X_d]^\top$, the sparse graph recovery problem aims to estimate its covariance matrix $\Sigma^*$ and precision matrix $\Theta^* = (\Sigma^*)^{-1}$. The $ij$-th component of $\Theta^*$ is zero if and only if $X_i$ and $X_j$ are conditionally independent given the other variables $\{X_k\}_{k \neq i,j}$. Therefore, it is popular to impose an $\ell_1$ regularization for the estimation of $\Theta^*$ to increase its sparsity and lead to easily interpretable models. Following Banerjee et al. (2008), the problem is formulated as the $\ell_1$-regularized maximum likelihood estimation

$$\widehat{\Theta} = \arg\min_{\Theta \in \mathcal{S}_{++}^d} \quad -\log(\det \Theta) + \text{tr}(\widehat{\Sigma}\Theta) + \rho \|\Theta\|_{1,\text{off}}, \tag{1}$$

where $\widehat{\Sigma}$ is the empirical covariance matrix based on $m$ samples, $\mathcal{S}_{++}^d$ is the space of $d \times d$ symmetric positive definite matrices (SPD), and $\|\Theta\|_{1,\text{off}} = \sum_{i \neq j} |\Theta_{ij}|$ is the off-diagonal $\ell_1$ regularizer with regularization parameter $\rho$. This estimator is sensible even for non-Gaussian $X$, since it is minimizing an $\ell_1$-penalized log-determinant Bregman divergence Ravikumar et al. (2011). The sparse precision matrix estimation problem in Eq. (1) is a convex optimization problem which can be solved by many algorithms. We give a few canonical and advanced examples which are compared in our experiments:

**G-ISTA.** G-ISTA is a proximal gradient method, and it updates the precision matrix iteratively

$$\Theta_{k+1} \leftarrow \eta_{\xi_k \rho}(\Theta_k - \xi_k(\widehat{\Sigma} - \Theta_k^{-1})), \quad \text{where } [\eta_\rho(X)]_{ij} := \text{sign}(X_{ij})(|X_{ij}| - \rho)_+. \tag{2}$$

The step sizes $\xi_k$ is determined by line search such that $\Theta_{k+1}$ is SPD matrix Rolfs et al. (2012).

**ADMM.** Alternating direction methods of multiplier (Boyd et al., 2011) transform the problem into an equivalent constrained form, decouple the log-determinant term and the $\ell_1$ regularization term, and result in the following augmented Lagrangian form with a penalty parameter $\lambda$:

$$- \log(\det \Theta) + \text{tr}(\widehat{\Sigma}\Theta) + \rho \|Z\|_1 + \langle \beta, \Theta - Z \rangle + \tfrac{1}{2}\lambda \|Z - \Theta\|_F^2. \tag{3}$$

Taking $U := \beta/\lambda$ as the scaled dual variable, the update rules for the ADMM algorithm are

$$\Theta_{k+1} \leftarrow \left( -Y + \sqrt{Y^\top Y + (4/\lambda)I} \right)/2, \quad \text{where } Y = \widehat{\Sigma}/\lambda - Z_k + U_k \tag{4}$$

$$Z_{k+1} \leftarrow \eta_{\rho/\lambda}(\Theta_{k+1} + U_k), \quad U_{k+1} \leftarrow U_k + \Theta_{k+1} - Z_{k+1} \tag{5}$$

**BCD.** Block-coordinate decent methods Friedman et al. (2008) updates each column (and the corresponding row) of the precision matrix iteratively by solving a sequence of lasso problems. The algorithm is very efficient for large scale problems involving thousands of variables.

Apart from various algorithms, rigorous statistical analysis has also been provided for the optimal solution of the convex formulation in Eq. (1). Ravikumar et al. (2011) established consistency of the estimator $\widehat{\Theta}$ in Eq. (1) in terms of both Frobenius and spectral norms, at rate scaling roughly as $\|\widehat{\Theta} - \Theta^*\| = \mathcal{O}\left( ((d+s)\log d/m)^{1/2} \right)$ with high probability, where $s$ is the number of nonzero entries in $\Theta^*$. This statistical analysis also reveal certain **limitations** of the convex formulation:

The established consistency is based on a set of **carefully chosen conditions**, including the lower bound of sample size, the sparsity level of $\Theta^*$, the degree of the graph, the magnitude of the entries in the covariance matrix, and the strength of interaction between edge and non-edge in the precision matrix (or mutual incoherence on the Hessian $\Gamma^* := \Sigma^* \otimes \Sigma^*$) . In practice, it may be hard to a problem to satisfy these recovery conditions.

Therefore, it seems that there is still room for improving the above convex optimization algorithms for recovering the true graph structure. Prior to the data-driven paradigm for sparse recovery, since the target parameter $\Theta^*$ is unknown, the best precision matrix recovery method is to resort to a surrogate objective function (for instance, equation 1). Optimally tuning the unknown parameter $\rho$ is a very challenging problem in practice. Instead, we can leverage the large amount of simulation or real data and design a learning algorithm that directly optimizes the loss in equation 9.

Furthermore, since the log-determinant estimator in Eq. (1) is NOT directly optimizing the recovery objective $\|\widehat{\Theta} - \Theta^*\|_F^2$, there is also a **mismatch** in the optimization objective and the final evaluation objective (refer to the first experiment in section 5.1). This increase the hope one may improve the results by directly optimizing the recovery objective with the algorithms learned from data.

## 3 LEARNING DATA-DRIVEN ALGORITHM FOR GRAPH RECOVERY

In the remainder of the paper, we will present a data-driven method to learn an algorithm for precision matrix estimation, and we call the resulting algorithm GLAD (stands for **G**raph recovery **L**earning **A**lgorithm using **D**ata-driven training). We ask the question of

> Given a family of precision matrices, is it possible to improve recovery results for sparse graphs by learning a data-driven algorithm?

More formally, suppose we are given $n$ precision matrices $\{\Theta^{*(i)}\}_{i=1}^n$ from a family $\mathcal{G}$ of graphs and $m$ samples $\{\boldsymbol{x}^{(i,j)}\}_{j=1}^m$ associated with each $\Theta^{*(i)}$. These samples can be used to form $n$ sample covariance matrices $\{\widehat{\Sigma}^{(i)}\}_{i=1}^n$. We are interested in learning an algorithm for precision matrix estimation by solving a supervised learning problem, $\min_f \frac{1}{n} \sum_{i=1}^n \mathcal{L}(\text{GLAD}_f(\widehat{\Sigma}^{(i)}), \Theta^{*(i)})$, where $f$ is a set of parameters in GLAD$(\cdot)$ and the output of GLAD$_f(\widehat{\Sigma}^{(i)})$ is expected to be a good estimation of $\Theta^{*(i)}$ in terms of an interested evaluation metric $\mathcal{L}$. The benefit is that it can directly optimize the final evaluation metric which is related to the desired structure or graph properties of a family of problems. However, it is a challenging task to design a good parameterization of GLAD$_f$ for this graph recovery problem. We will explain the challenges below and then present our solution.

### 3.1 CHALLENGES IN DESIGNING LEARNING MODELS

In the literature on learning data-driven algorithms, most models are designed using traditional deep learning architectures, such as fully connected DNN or recurrent neural networks. But, for graph recovery problems, directly using these architectures does not work well due to the following reasons.

First, using a fully connected neural network is not practical. Since both the input and the output of graph recovery problems are matrices, the number of parameters scales at least quadratically in $d$. Such a large number of parameters will need many input-output training pairs to provide a decent estimation. Thus some structures need to be imposed in the network to reduce the size of parameters and sample complexity.

Second, structured models such as convolution neural networks (CNNs) have been applied to learn a mapping from $\widehat{\Sigma}$ to $\Theta^*$ (Belilovsky et al., 2017). Due to the structure of CNNs, the number of parameters can be much smaller than fully connected networks. However, a recovered graph should be permutation invariant with respect to the matrix rows/columns, and this constraint is very hard to be learned by CNNs, unless there are lots of samples. Also, the structure of CNN is a bias imposed on the model, and there is no guarantee why this structure may work.

Third, the intermediate results produced by both fully connected networks and CNNs are not interpretable, making it hard to diagnose the learned procedures and progressively output increasingly improved precision matrix estimators.

Fourth, the SPD constraint is hard to impose in traditional deep learning architectures.

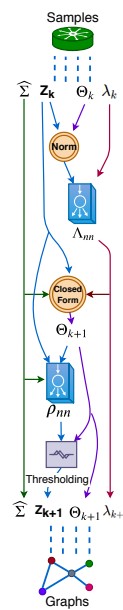

Figure 1: A recurrent unit `GLADcell`.

Although, the above limitations do suggest a list of desiderata when designing learning models: Small model size; Minimalist learning; Interpretable architecture; Progressive improvement; and SPD output. These desiderata will motivate the design of our deep architecture using unrolled algorithms.

### 3.2 GLAD: DEEP LEARNING MODEL BASED ON UNROLLED ALGORITHM

To take into account the above desiderata, we will use an unrolled algorithm as the template for the architecture design of GLAD. The unrolled algorithm already incorporates some problem structures, such as permutation invariance and interpretable intermediate results; but this unrolled algorithm does not traditionally have a learning component, and is typically not directly suitable for gradient-based approaches. We will leverage this inductive bias in our architecture design and augment the unrolled algorithm with suitable and flexible learning components, and then train these embedded models with stochastic gradient descent.

GLAD model is based on a reformulation of the original optimization problem in Eq. (1) with a squared penalty term, and an alternating minimization (AM) algorithm for it. More specifically, we consider a modified optimization with a quadratic penalty parameter $\lambda$:

$$\widehat{\Theta}_\lambda, \widehat{Z}_\lambda := \arg\min_{\Theta, Z \in \mathcal{S}_{++}^d} -\log(\det \Theta) + \mathrm{tr}(\widehat{\Sigma}\Theta) + \rho\|Z\|_1 + \tfrac{1}{2}\lambda\|Z - \Theta\|_F^2 \qquad (6)$$

and the alternating minimization (AM) method for solving it:

$$\Theta_{k+1}^{\mathrm{AM}} \leftarrow \tfrac{1}{2}\Big(-Y + \sqrt{Y^\top Y + \tfrac{4}{\lambda}I}\Big), \text{ where } Y = \tfrac{1}{\lambda}\widehat{\Sigma} - Z_k^{\mathrm{AM}}; \qquad (7)$$

$$Z_{k+1}^{\mathrm{AM}} \leftarrow \eta_{\rho/\lambda}(\Theta_{k+1}^{\mathrm{AM}}), \qquad (8)$$

where $\eta_{\rho/\lambda}(\theta) := \mathrm{sign}(\theta)\max(|\theta| - \rho/\lambda, 0)$. The derivation of these steps are given in Appendix A. We replace the penalty constants $(\rho, \lambda)$ by problem dependent neural networks, $\rho_{nn}$ and $\Lambda_{nn}$. These neural networks are minimalist in terms of the number of parameters as the input dimensions are mere $\{3, 2\}$ for $\{\rho_{nn}, \Lambda_{nn}\}$ and outputs a single value. Algorithm 1 summarizes the update equations for our unrolled AM based model, GLAD. Except for the parameters in $\rho_{nn}$ and $\Lambda_{nn}$, the constant $t$ for initialization is also a learnable scalar parameter. This unrolled algorithm with neural network augmentation can be viewed as a highly structured recurrent architecture as illustrated in Figure 1.

There are many traditional algorithms for solving graph recovery problems. We choose AM as our basis because: First, empirically, we tried models built upon other algorithms including G-ISTA, ADMM, etc, but AM-based model gives consistently better performances. Appendix C.10 & C.11

discusses different parameterizations tried. Second, and more importantly, the AM-based architecture has a nice property of maintaining $\Theta_{k+1}$ as a SPD matrix throughout the iterations as long as $\lambda_k < \infty$. Third, as we prove later in Section 4, the AM algorithm has linear convergence rate, allowing us to use a fixed small number of iterations and still achieve small error margins.

### 3.3 TRAINING ALGORITHM

To learn the parameters in GLAD architecture, we will directly optimize the recovery objective function rather than using log-determinant objective. A nice property of our deep learning architecture is that each iteration of our model will output a valid precision matrix estimation. This allows us to add auxiliary losses to regularize the intermediate results of our GLAD architecture, guiding it to learn parameters which can generate a smooth solution trajectory.

Specifically, we will use Frobenius norm in our experiments, and design an objective which has some resemblance to the discounted cumulative reward in reinforcement learning:

$$\min_{f} \; \text{loss}_f := \frac{1}{n} \sum_{i=1}^{n} \sum_{k=1}^{K} \gamma^{K-k} \left\| \Theta_k^{(i)} - \Theta^* \right\|_F^2, \quad (9)$$

where $(\Theta_k^{(i)}, Z_k^{(i)}, \lambda_k^{(i)}) = \texttt{GLADcell}_f(\widehat{\Sigma}^{(i)}, \Theta_{k-1}^{(i)}, Z_{k-1}^{(i)}, \lambda_{k-1}^{(i)})$ is the output of the recurrent unit GLADcell at $k$-th iteration, $K$ is number of unrolled iterations, and $\gamma \leq 1$ is a discounting factor.

---

**Algorithm 1:** GLAD

**Function** GLADcell $(\widehat{\Sigma}, \Theta, Z, \lambda)$:

  $\lambda \leftarrow \Lambda_{nn}(\|Z - \Theta\|_F^2, \lambda)$

  $Y \leftarrow \lambda^{-1}\widehat{\Sigma} - Z$

  $\Theta \leftarrow \frac{1}{2}\left( -Y + \sqrt{Y^\top Y + \frac{4}{\lambda}I} \right)$

  **For** *all* $i, j$ **do**

    $\rho_{ij} = \rho_{nn}(\Theta_{ij}, \widehat{\Sigma}_{ij}, Z_{ij})$

    $Z_{ij} \leftarrow \eta_{\rho_{ij}}(\Theta_{ij})$

  **return** $\Theta, Z, \lambda$

**Function** GLAD $(\widehat{\Sigma})$:

  $\Theta_0 \leftarrow (\widehat{\Sigma} + tI)^{-1}, \lambda_0 \leftarrow 1$

  **For** $k = 0$ *to* $K - 1$ **do**

    $\Theta_{k+1}, Z_{k+1}, \lambda_{k+1}$

    $\leftarrow$GLADcell $(\widehat{\Sigma}, \Theta_k, Z_k, \lambda_k)$

  **return** $\Theta_K, Z_K$

---

We will use stochastic gradient descent algorithm to train the parameters $f$ in the GLADcell. A key step in the gradient computation is to propagate gradient through the matrix square root in the GLADcell. To do this efficiently, we make use of the property of SPD matrix that $X = X^{1/2}X^{1/2}$, and the product rule of derivatives to obtain

$$dX = d(X^{1/2})X^{1/2} + X^{1/2}d(X^{1/2}). \quad (10)$$

The above equation is a Sylvester's equation for $d(X^{1/2})$. Since the derivative $dX$ for $X$ is easy to obtain, then the derivative of $d(X^{1/2})$ can be obtained by solving the Sylvester's equation in (10).

The objective function in equation 9 should be understood in a similar way as in Gregor & LeCun (2010); Belilovsky et al. (2017); Liu et al. (2018) where deep architectures are designed to directly produce the sparse outputs.

For GLAD architecture, a collection of input covariance matrix and ground truth sparse precision matrix pairs are available during training, either coming from simulated or real data. Thus the objective function in equation 9 is formed to directly compare the output of GLAD with the ground truth precision matrix. The goal is to train the deep architecture which can perform well for a family/distribution of input covariance matrix and ground truth sparse precision matrix pairs. The average in the objective function is over different input covariance and precision matrix pairs such that the learned architecture is able to perform well over a family of problem instances.

Furthermore, each layer of our deep architecture outputs an intermediate prediction of the sparse precision matrix. The objective function takes into account all these intermediate outputs, weights the loss according to the layer of the deep architecture, and tries to progressively bring these intermediate layer outputs closer and closer to the target ground truth.

### 3.4 A NOTE ON GLAD ARCHITECTURE'S EXPRESSIVE ABILITY

We note that the designed architecture, is more flexible than just learning the regularization parameters. The component in GLAD architecture corresponding to the regularization parameters are entry-wise and also adaptive to the input covariance matrix and the intermediate outputs. GLAD architecture can adaptively choose a matrix of regularization parameters. This task will be very challenging if the matrix of regularization parameters are tuned manually using cross-validation. A recent theoretical work Sun et al. (2018) also validates the choice of GLAD's design.

## 4 THEORETICAL ANALYSIS

Since GLAD architecture is obtained by augmenting an unrolled optimization algorithm by learnable components, the question is what kind of guarantees can be provided for such learned algorithm, and whether learning can bring benefits to the recovery of the precision matrix. In this section, we will first analyze the statistical guarantee of running the AM algorithm in Eq. (7) and Eq. (8) for $k$ steps with a fixed quadratic penalty parameter $\lambda$, and then interpret its implication for the learned algorithm. First, we need some standard assumptions about the true model from the literature Rothman et al. (2008):

**Assumption 1.** *Let the set $S = \{(i,j) : \Theta_{ij}^* \neq 0, i \neq j\}$. Then $card(S) \leq s$.*

**Assumption 2.** $\Lambda_{\min}(\Sigma^*) \geq \epsilon_1 > 0$ *(or equivalently $\Lambda_{\max}(\Theta^*) \leq 1/\epsilon_1$), $\Lambda_{\max}(\Sigma^*) \leq \epsilon_2$ and an upper bound on $\|\widehat{\Sigma}\|_2 \leq c_{\widehat{\Sigma}}$.*

The assumption 2 guarantees that $\Theta^*$ exists. Assumption 1 just upper bounds the sparsity of $\Theta^*$ and does not stipulate anything in particular about $s$. These assumptions characterize the fundamental limitation of the sparse graph recovery problem, beyond which recovery is not possible. Under these assumptions, we prove the linear convergence of AM algorithm (proof is in Appendix B).

**Theorem 1.** *Under the assumptions 1 & 2, if $\rho \asymp \sqrt{\frac{\log d}{m}}$, where $\rho$ is the $l_1$ penalty, $d$ is the dimension of problem and $m$ is the number of samples, the Alternate Minimization algorithm has linear convergence rate for optimization objective defined in (6). The $k^{th}$ iteration of the AM algorithm satisfies,*

$$\left\|\Theta_k^{AM} - \Theta^*\right\|_F \leqslant C_\lambda \left\|\Theta_{k-1}^{AM} - \widehat{\Theta}_\lambda\right\|_F + \mathcal{O}_{\mathbb{P}}\left(\sqrt{\frac{(\log d)/m}{\min(\frac{1}{(d+s)}, \frac{\lambda}{d^2})}}\right), \tag{11}$$

*where $0 < C_\lambda < 1$ is a constant depending on $\lambda$.*

From the theorem, one can see that by optimizing the quadratic penalty parameter $\lambda$, one can adjust the $C_\lambda$ in the bound. We observe that at each stage $k$, an optimal penalty parameter $\lambda_k$ can be chosen depending on the most updated value $C_\lambda$. An adaptive sequence of penalty parameters $(\lambda_1, \ldots, \lambda_K)$ should achieve a better error bound compared to a fixed $\lambda$. Since $C_\lambda$ is a very complicated function of $\lambda$, the optimal $\lambda_k$ is hard to choose manually.

Besides, the linear convergence guarantee in this theorem is based on the sparse regularity parameter $\rho \asymp \sqrt{\frac{\log d}{m}}$. However, choosing a good $\rho$ value in practice is tedious task as shown in our experiments.

In summary, the implications of this theorem are:

- An adaptive sequence $(\lambda_1, \ldots, \lambda_K)$ should lead to an algorithm with better convergence than a fixed $\lambda$, but the sequence may not be easy to choose manually.
- Both $\rho$ and the optimal $\lambda_k$ depend on the corresponding error $\|\Theta^{AM} - \widehat{\Theta}_\lambda\|_F$, which make these parameters hard to prescribe manually.
- Since, the AM algorithm has a fast linear convergence rate, we can run it for a fixed number of iterations $K$ and still converge with a reasonable error margin.

Our learning augmented deep architecture, GLAD, can tune these sequence of $\lambda_k$ and $\rho$ parameters jointly using gradient descent. Moreover, we refer to a recent work by Sun et al. (2018) where they considered minimizing the graphical lasso objective with a general nonconvex penalty. They showed that by iteratively solving a sequence of adaptive convex programs one can achieve even better error margins (refer their Algorithm 1 & Theorem 3.5). In every iteration they chose an adaptive regularization matrix based on the most recent solution and the choice of nonconvex penalty. We thus hypothesize that we can further improve our error margin if we make the penalty parameter $\rho$ nonconvex and problem dependent function. We choose $\rho$ as a function depending on the most up-to-date solution $(\Theta_k, \widehat{\Sigma}, Z_k)$, and allow different regularizations for different entries of the precision matrix. Such flexibility potentially improves the ability of GLAD model to recover the sparse graph.

## 5 EXPERIMENTS

In this section, we report several experiments to compare GLAD with traditional algorithms and other data-driven algorithms. The results validate the list of desiderata mentioned previously. Especially, it shows the potential of pushing the boundary of traditional graph recovery algorithms by utilizing data. Python implementation (tested on P100 GPU) is available[1]. Exact experimental settings details are covered in Appendix C. **Evaluation metric.** We use normalized mean square error (NMSE) and probability of success (PS) to evaluate the algorithm performance. NMSE is $10 \log_{10}(\mathbb{E} \|\Theta^p - \Theta^*\|_F^2 / \mathbb{E} \|\Theta^*\|_F^2)$ and PS is the probability of correct signed edge-set recovery, i.e., $\mathbb{P}\left[\text{sign}(\Theta_{ij}^p) = \text{sign}(\Theta_{ij}^*), \forall (i,j) \in \mathbf{E}(\Theta^*)\right]$, where $\mathbf{E}(\Theta^*)$ is the true edge set. **Notation.** In all reported results, D stands for dimension $d$ of the random variable, M stands for sample size and N stands for the number of graphs (precision matrices) that is used for training.

### 5.1 BENEFIT OF DATA-DRIVEN GRADIENT-BASED ALGORITHM

**Inconsistent optimization objective.** Traditional algorithms are typically designed to optimize the $\ell_1$-penalized log likelihood. Since it is a convex optimization, convergence to optimal solution is usually guaranteed. However, this optimization objective is different from the true error. Taking ADMM as an example, it is revealed in Figure 2 that, although the optimization objective always converges, errors of recovering true precision matrices measured by NMSE have very different behaviors given different regularity parameter $\rho$, which indicates the necessity of directly optimizing NMSE and hyperparameter tuning.

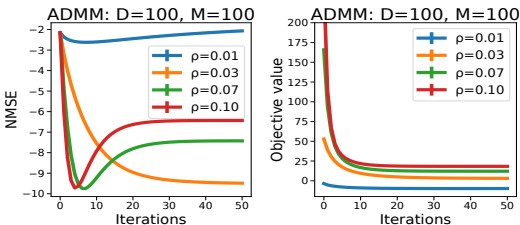

Figure 2: Convergence of ADMM in terms of NMSE and optimization objective. (Refer to Appendix C.2).

**Expensive hyperparameter tuning.** Although hyperparameters of traditional algorithms can be tuned if the true precision matrices are provided as a validation dataset, we want to emphasize that hyperparamter tuning by **grid search** is a tedious and hard task. Table 1 shows that the NMSE values are very sensitive to both $\rho$ and the quadratic penalty $\lambda$ of ADMM method. For instance, the optimal

| $\rho$ \ $\lambda$ | 5 | 1 | 0.5 | **0.1** | 0.01 |
|---|---|---|---|---|---|
| 0.01 | -2.51 | -2.25 | -2.06 | -2.06 | -2.69 |
| **0.03** | -5.59 | -9.05 | 9.48 | **-9.61** | -9.41 |
| 0.07 | -9.53 | -7.58 | -7.42 | -7.38 | -7.46 |
| 0.1 | -9.38 | -6.51 | -6.43 | -6.41 | -6.50 |
| 0.2 | -6.76 | -4.68 | -4.55 | -4.47 | -4.80 |

Table 1: NMSE results for ADMM.

NMSE in this table is $-9.61$ when $\lambda = 0.1$ and $\rho = 0.03$. However, it will increase by a large amount to $-2.06$ if $\rho$ is only changed slightly to $0.01$. There are many other similar observations in this table, where slight changes in parameters can lead to significant NMSE differences, which in turns makes grid-search very expensive. G-ISTA and BCD follow similar trends.

For a fair comparison against GLAD which is data-driven, in all following experiments, all hyperparameters in traditional algorithms are **fine-tuned** using validation datasets, for which we spent extensive efforts (See more details in Appendix C.3, C.6). In contrast, the gradient-based training of GLAD turns out to be much easier.

### 5.2 CONVERGENCE

We follow the experimental setting in (Rolfs et al., 2012; Mazumder & Agarwal, 2011; Lu, 2010) to generate data and perform synthetic experiments on multivariate Gaussians. Each off-diagonal entry of the precision matrix is drawn from a uniform distribution, i.e., $\Theta_{ij}^* \sim \mathcal{U}(-1, 1)$, and then set to zero with probability $p = 1 - s$, where $s$ means the sparsity level. Finally, an appropriate multiple

$$\lambda \leftarrow \Lambda_{nn}(\|Z - \Theta\|_F^2, \lambda) \quad \rho_{ij} = \rho_{nn}(\Theta_{ij}, \widehat{\Sigma}_{ij}, Z_{ij})$$

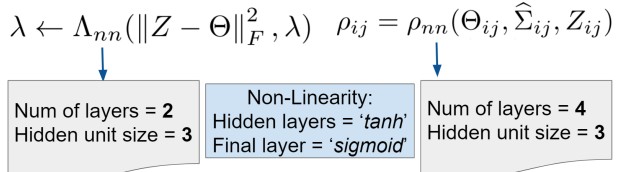

Figure 3: **Minimalist** neural network architectures designed for GLAD experiments in sections(5.2, 5.3, 5.4, 5.5). Refer Appendix C.5 for further details about the architecture parameters.

of the identity matrix was added to the current matrix, so that the resulting matrix had the smallest eigenvalue as 1 (refer to Appendix C.1). We use 30 unrolled steps for GLAD (Figure 3) and compare

---

[1]code: https://drive.google.com/open?id=16POE4TMp7UUieLcLqRzSTqzkVHm2stlM

it to G-ISTA, ADMM and BCD. All algorithms are trained/finetuned using 10 randomly generated graphs and tested over 100 graphs.

Convergence results and average runtime of different algorithms on Nvidia's P100 GPUs are shown in Figure 4 and Table 2 respectively. `GLAD` consistently converges faster and gives lower NMSE. Although the fine-tuned G-ISTA also has decent performance, the computation time in each iteration is much longer than `GLAD` because it requires line search steps. Besides, we could also see a progressive improvement of `GLAD` across its iterations.

| Time/itr | D=25 | D=100 |
|----------|------|-------|
| ADMM     | 1.45 | 16.45 |
| G-ISTA   | 37.51| 41.47 |
| GLAD     | 2.81 | 20.23 |

Table 2: ms per iteration.

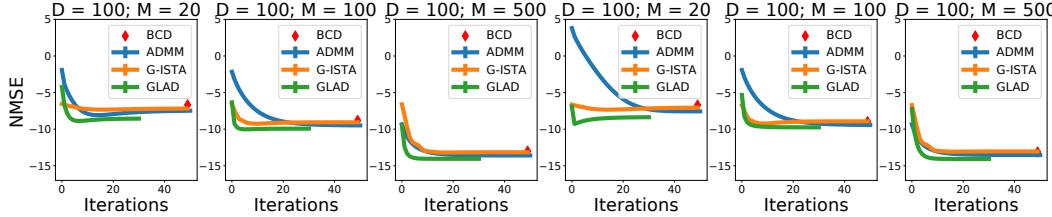

Figure 4: GLAD vs traditional methods. *Left 3 plots:* Sparsity level is fixed as $s = 0.1$. *Right 3 plots:* Sparsity level of each graph is randomly sampled as $s \sim \mathcal{U}(0.05, 0.15)$. Results are averaged over 100 test graphs where each graph is estimated 10 times using 10 different sample batches of size $M$. Standard error is plotted but not visible. Intermediate steps of BCD are not evaluated because we use sklearn packagePedregosa et al. (2011) and can only access the final output. Appendix C.4, C.5 explains the experiment setup and `GLAD` architecture.

## 5.3 RECOVERY PROBABILITY

As analyzed by Ravikumar et al. (2011), the recovery guarantee (such as in terms of Frobenius norm) of the $\ell_1$ regularized log-determinant optimization significantly depends on the sample size and other conditions. Our `GLAD` directly optimizes the recovery objective based on data, and it has the potential of pushing the sample complexity limit. We experimented with this and found the results positive.

We follow Ravikumar et al. (2011) to conduct experiments on GRID graphs, which satisfy the conditions required in (Ravikumar et al., 2011). Furthermore, we conduct a more challenging task of recovering restricted but randomly constructed graphs (see Appendix C.7 for more details). The probability of success (PS) is

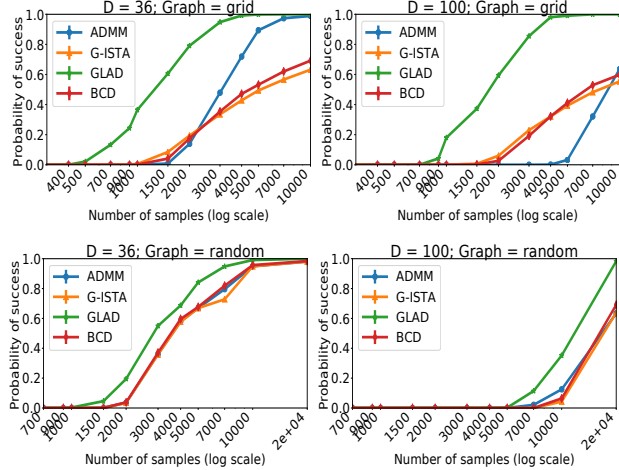

Figure 5: Sample complexity for model selection consistency.

non-zero only if the algorithm recovers all the edges with correct signs, plotted in Figure 5. `GLAD` consistently outperforms traditional methods in terms of sample complexity as it recovers the true edges with considerably fewer number of samples.

## 5.4 DATA EFFICIENCY

Having a good inductive bias makes `GLAD`'s architecture quite data-efficient compared to other deep learning models. For instance, the state-of-the-art 'DeepGraph' by Belilovsky et al. (2017) is based on CNNs. It contains orders of magnitude more parameters than `GLAD`. Furthermore, it takes roughly 100,000 samples, and several hours for training their DG-39 model. In contrast, `GLAD` learns well with less than 25 parameters, within 100 training samples, and notably less training time. Table 3 also shows that `GLAD` significantly outperforms DG-39 model in terms of AUC (Area under the ROC curve) by just using 100 training graphs, typically the case for real world settings. Fully connected DL models are unable to learn from such small data and hence are skipped in the comparison.

Table 3: AUC on 100 test graphs for D=39: For experiment settings, refer Table 1 of Belilovsky et al. (2017). Gaussian Random graphs with sparsity $p = 0.05$ were chosen and edge values sampled from $\sim \mathcal{U}(-1, 1)$. (Refer appendix(C.8))

| Methods | M=15 | M=35 | M=100 |
|---------|------|------|-------|
| BCD | 0.578±0.006 | 0.639±0.007 | 0.704±0.006 |
| DG-39 | 0.664±0.008 | 0.738±0.006 | 0.759±0.006 |
| DG-39+P | 0.672±0.008 | 0.740±0.007 | 0.771±0.006 |
| GLAD | **0.788±0.003** | **0.811±0.003** | **0.878±0.003** |

## 5.5 GENE REGULATION DATA

The SynTReN (Van den Bulcke et al., 2006) is a synthetic gene expression data generator specifically designed for analyzing the sparse graph recovery algorithms. It models different types of biological interactions and produces biologically plausible synthetic gene expression data. Figure 6 shows that GLAD performs favourably for structure recovery in terms of NMSE on the gene expression data. As the governing equations of the underlying distribution of the SynTReN are unknown, these experiments also emphasize the ability of GLAD to handle non-Gaussian data.

Figure 7 visualizes the edge-recovery performance of GLAD models trained on a sub-network of true Ecoli bacteria data. We denote, TPR: True Positive Rate, FPR: False Positive Rate, FDR: False Discovery Rate. The number of simulated training/validation graphs were set to 20/20. One batch of $M$ samples were taken per graph (details in **Appendix C.9**). Although, GLAD was trained on graphs with $D = 25$, it was able to robustly recover a higher dimensional graph $D = 43$ structure.

Appendix C.12 contains details of the experiments done on real E.Coli data. The GLAD model was trained using the SynTReN simulator.

Appendix C.13 explains our proposed approach to scale for larger problem sizes.

Figure 6: Performance on the SynTReN generated gene expression data with graph as Erdos-renyi having sparsity $p = 0.05$. Refer appendix(C.9) for experiment details.

Figure 7: Recovered graph structures for a sub-network of the *E. coli* consisting of 43 genes and 30 interactions with increasing samples. Increasing the samples reduces the fdr by discovering more true edges.

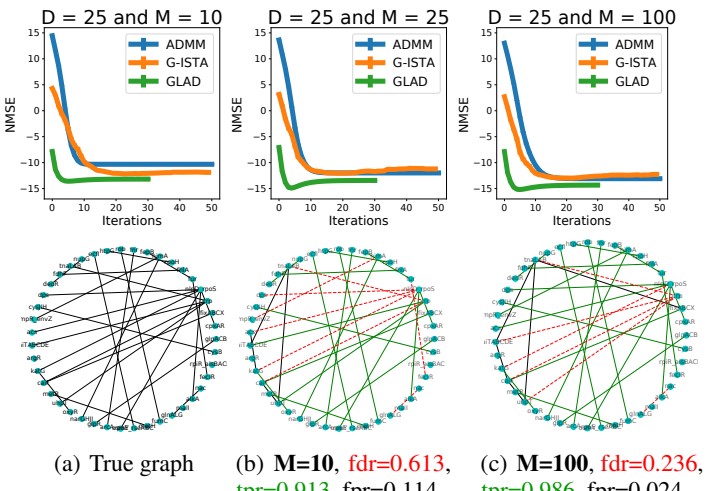

(a) True graph    (b) **M=10**, fdr=0.613, tpr=0.913, fpr=0.114    (c) **M=100**, fdr=0.236, tpr=0.986, fpr=0.024

## 6 CONCLUSION & FUTURE WORK

We presented a novel neural network, GLAD, for the sparse graph recovery problem based on an unrolled Alternating Minimization algorithm. We theoretically prove the linear convergence of AM algorithm as well as empirically show that learning can further improve the sparse graph recovery. The learned GLAD model is able to push the sample complexity limits thereby highlighting the potential of using algorithms as inductive biases for deep learning architectures. Further development of theory is needed to fully understand and realize the potential of this new direction.

## ACKNOWLEDGEMENT

We thank our colleague Haoran Sun for his helpful comments. This research was supported in part through research cyberinfrastructure resources and services provided by the Partnership for an Advanced Computing Environment (PACE) at the Georgia Institute of Technology, Atlanta, Georgia, USA (PACE, 2017). This research was also partly supported by XSEDE Campus Champion Grant GEO150002.

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

# APPENDIX

## A   DERIVATION OF ALTERNATING MINIMIZATION STEPS

Given the optimization problem

$$\widehat{\Theta}_\lambda, \widehat{Z}_\lambda := \arg\min_{\Theta, Z \in \mathcal{S}_{++}^d} -\log(\det \Theta) + \operatorname{tr}(\widehat{\Sigma}\Theta) + \rho \|Z\|_1 + \tfrac{1}{2}\lambda \|Z - \Theta\|_F^2, \quad (12)$$

Alternating Minimization is performing

$$\Theta_{k+1}^{AM} \leftarrow \arg\min_{\Theta \in \mathcal{S}_{++}^d} -\log(\det \Theta) + \operatorname{tr}(\widehat{\Sigma}\Theta) + \tfrac{1}{2}\lambda \left\|Z_k^{AM} - \Theta\right\|_F^2 \quad (13)$$

$$Z_{k+1}^{AM} \leftarrow \arg\min_{Z \in \mathcal{S}_{++}^d} \operatorname{tr}(\widehat{\Sigma}\Theta_{k+1}^{AM}) + \rho \|Z\|_1 + \tfrac{1}{2}\lambda \left\|Z - \Theta_{k+1}^{AM}\right\|_F^2. \quad (14)$$

Taking the gradient of the objective function with respect to $\Theta$ to be zero, we have

$$-\Theta^{-1} + \widehat{\Sigma} + \lambda(\Theta - Z) = 0. \quad (15)$$

Taking the gradient of the objective function with respect to $Z$ to be zero, we have

$$\rho \partial \ell_1(Z) + \lambda(Z - \Theta) = 0, \quad (16)$$

where

$$\partial \ell_1(Z_{ij}) = \begin{cases} 1 & Z_{ij} > 0, \\ -1 & Z_{ij} < 0, \\ [-1, 1] & Z_{ij} = 0. \end{cases} \quad (17)$$

Solving the above two equations, we obtain:

$$\frac{1}{2}\left(-Y + \sqrt{Y^\top Y + \frac{4}{\lambda}I}\right) = \arg\min_{\Theta \in \mathcal{S}_{++}^d} -\log(\det \Theta) + \operatorname{tr}(\widehat{\Sigma}\Theta) + \frac{1}{2}\lambda \|Z - \Theta\|_F^2, \quad (18)$$

$$\text{where } Y = \frac{1}{\lambda}\widehat{\Sigma} - Z, \quad (19)$$

$$\eta_{\rho/\lambda}(\Theta) = \arg\min_{Z \in \mathcal{S}_{++}^d} \operatorname{tr}(\widehat{\Sigma}\Theta) + \rho \|Z\|_1 + \frac{1}{2}\lambda \|Z - \Theta\|_F^2. \quad (20)$$

## B   LINEAR CONVERGENCE RATE ANALYSIS

**Proof of Theorem**

**Theorem 1.** *Under the assumptions 1 & 2, if $\rho \asymp \sqrt{\frac{\log d}{m}}$, where $\rho$ is the $l_1$ penalty, $d$ is the dimension of problem and $m$ is the number of samples, the Alternate Minimization algorithm has linear convergence rate for optimization objective defined in (6). The $k^{th}$ iteration of the AM algorithm satisfies,*

$$\left\|\Theta_k^{AM} - \Theta^*\right\|_F \leqslant C_\lambda \left\|\Theta_{k-1}^{AM} - \widehat{\Theta}_\lambda\right\|_F + \mathcal{O}_\mathbb{P}\left(\sqrt{\frac{(\log d)/m}{\min(\frac{1}{(d+s)}, \frac{\lambda}{d^2})}}\right), \quad (11)$$

*where $0 < C_\lambda < 1$ is a constant depending on $\lambda$.*

We will reuse the following notations in the appendix:

$$\widehat{\Sigma}^m : \text{ sample covariance matrix based on } m \text{ samples}, \tag{21}$$

$$\mathcal{G}(\Theta; \rho) := -\log(\det \Theta) + \text{tr}(\hat{\Sigma}^m \Theta) + \rho \|\Theta\|_{1,\text{off}}, \tag{22}$$

$$\widehat{\Theta}_\mathcal{G} := \arg\min_{\Theta \in \mathcal{S}_{++}^d} \quad \mathcal{G}(\Theta; \rho), \tag{23}$$

$$f(\Theta, Z; \rho, \lambda) := -\log(\det \Theta) + \text{tr}(\widehat{\Sigma}\Theta) + \rho \|Z\|_1 + \frac{1}{2}\lambda \|Z - \Theta\|_F^2, \tag{24}$$

$$\widehat{\Theta}_\lambda, \widehat{Z}_\lambda := \arg\min_{\theta, Z \in \mathcal{S}_{++}^d} f(\Theta, Z; \rho, \lambda), \tag{25}$$

$$f^*(\rho, \lambda) := \min_{\theta, Z \in \mathcal{S}_{++}^d} f(\Theta, Z; \rho, \lambda) = f(\widehat{\Theta}_\lambda, \widehat{Z}_\lambda; \rho, \lambda), \tag{26}$$

$$\eta_{\rho/\lambda}(\theta) := \text{sign}(\theta) \max(|\theta| - \rho/\lambda, 0). \tag{27}$$

The update rules for Alternating Minimization are:

$$\Theta_{k+1}^{\text{AM}} \leftarrow \frac{1}{2}\left(-Y + \sqrt{Y^\top Y + \frac{4}{\lambda}I}\right), \text{ where } Y = \frac{1}{\lambda}\widehat{\Sigma} - Z_k^{\text{AM}}; \tag{28}$$

$$Z_{k+1}^{\text{AM}} \leftarrow \eta_{\rho/\lambda}(\Theta_{k+1}^{\text{AM}}), \tag{29}$$

**Assumptions:** With reference to the theory developed in Rothman et al. (2008), we make the following assumptions about the true model. ($\mathcal{O}_\mathbb{P}(\cdot)$ is used to denote bounded in probability.)

**Assumption 1.** *Let the set* $S = \{(i, j) : \Theta_{ij}^* \neq 0, i \neq j\}$. *Then* $card(S) \leq s$.

**Assumption 2.** $\Lambda_{\min}(\Sigma^*) \geq \epsilon_1 > 0$ *(or equivalently* $\Lambda_{\max}(\Theta^*) \leq 1/\epsilon_1$*),* $\Lambda_{\max}(\Sigma^*) \leq \epsilon_2$ *and an upper bound on* $\|\widehat{\Sigma}\|_2 \leq c_{\widehat{\Sigma}}$.

We now proceed towards the proof:

**Lemma 2.** *For any* $x, y, k \in \mathbb{R}$, $k > 0$, $x \neq y$,

$$\frac{(\sqrt{x^2 + k} - \sqrt{y^2 + k})^2}{(x - y)^2} \leq 1 - \frac{1}{\sqrt{(\frac{x^2}{k} + 1)(\frac{y^2}{k} + 1)}}. \tag{30}$$

*Proof.*

$$\frac{(\sqrt{x^2 + k} - \sqrt{y^2 + k})^2}{(x - y)^2} = \frac{(x^2 + k) + (y^2 + k) - 2\sqrt{x^2 + k}\sqrt{y^2 + k}}{(x - y)^2} \tag{31}$$

$$= \frac{(x - y)^2 - 2(\sqrt{x^2 + k}\sqrt{y^2 + k} - (xy + k))}{(x - y)^2} \tag{32}$$

$$= 1 - 2\frac{(\sqrt{x^2 + k}\sqrt{y^2 + k} - (xy + k))(\sqrt{x^2 + k}\sqrt{y^2 + k} + (xy + k))}{(x - y)^2(\sqrt{x^2 + k}\sqrt{y^2 + k} + (xy + k))} \tag{33}$$

$$= 1 - 2\frac{k(x - y)^2}{(x - y)^2(\sqrt{x^2 + k}\sqrt{y^2 + k} + (xy + k))} \tag{34}$$

$$= 1 - \frac{2k}{\sqrt{x^2 + k}\sqrt{y^2 + k} + (xy + k)} \tag{35}$$

$$\leq 1 - \frac{1}{\sqrt{(\frac{x^2}{k} + 1)(\frac{y^2}{k} + 1)}} \tag{36}$$

$$\square$$

**Lemma 3.** *For any* $X, Y \in \mathcal{S}^d$, $\lambda > 0$, $A(Y) = \sqrt{Y^\top Y + \frac{4}{\lambda}I}$ *satisfies the following inequality,*

$$\|A(X) - A(Y)\|_F \leq \alpha_\lambda \|X - Y\|_F, \tag{37}$$

*where* $0 < \alpha_\lambda = 1 - \frac{1}{2}(\frac{\lambda}{4}\Lambda_{max}(X)^2 + 1)^{-1/2}(\frac{\lambda}{4}\Lambda_{max}(Y)^2 + 1)^{-1/2} < 1$, $\Lambda_{max}(X)$ *is the largest eigenvalue of* $X$ *in absolute value.*

*Proof.* First we factorize $X$ using eigen decomposition, $X = Q_X^\top D_X Q_X$, where $Q_X$ and $D_X$ are orthogonal matrix and diagonal matrix, respectively. Then we have,

$$A(X) = \sqrt{Q_X^\top D_X^2 Q_X + \frac{4}{\lambda} I} = \sqrt{Q_X^\top (D_X^2 + \frac{4}{\lambda} I) Q_X} = Q_X^\top \sqrt{D_X^2 + \frac{4}{\lambda} I} Q_X. \tag{38}$$

Similarly, the above equation holds for $Y$. Therefore,

$$\|A(X) - A(Y)\|_F = \left\| Q_X^\top \sqrt{D_X^2 + \frac{4}{\lambda} I} Q_X - Q_Y^\top \sqrt{D_Y^2 + \frac{4}{\lambda} I} Q_Y \right\|_F \tag{39}$$

$$= \left\| Q_X (Q_X^\top \sqrt{D_X^2 + \frac{4}{\lambda} I} Q_X - Q_Y^\top \sqrt{D_Y^2 + \frac{4}{\lambda} I} Q_Y) Q_X^\top \right\|_F \tag{40}$$

$$= \left\| \sqrt{D_X^2 + \frac{4}{\lambda} I} - Q_X Q_Y^\top \sqrt{D_Y^2 + \frac{4}{\lambda} I} Q_Y Q_X^\top \right\|_F \tag{41}$$

$$= \left\| \sqrt{D_X^2 + \frac{4}{\lambda} I} - Q^\top \sqrt{D_Y^2 + \frac{4}{\lambda} I} Q \right\|_F, \tag{42}$$

where we define $Q := Q_Y Q_X^\top$. Similarly, we have,

$$\|X - Y\|_F = \left\| Q_X^\top D_X Q_X - Q_Y^\top D_Y Q_Y \right\|_F \tag{43}$$

$$= \left\| D_X - Q_X Q_Y^\top D_Y Q_Y Q_X^\top \right\|_F \tag{44}$$

$$= \left\| D_X - Q^\top D_Y Q \right\|_F. \tag{45}$$

Then the $i$-th entry on the diagonal of $Q^\top D_Y Q$ is $\sum_{j=1}^d D_{Yjj} Q_{ji}^2$. Using the fact that $D_X$ and $D_Y$ are diagonal, we have,

$$\|X - Y\|_F^2 = \left\| D_X - Q^\top D_Y Q \right\|_F^2 \tag{46}$$

$$= \left\| Q^\top D_Y Q \right\|_F^2 - \sum_{i=1}^d (\sum_{j=1}^d D_{Yjj} Q_{ji}^2)^2 + \sum_{i=1}^d (D_{Xii} - \sum_{j=1}^d D_{Yjj} Q_{ji}^2)^2 \tag{47}$$

$$= \|D_Y\|_F^2 + \sum_{i=1}^d D_{Xii}(D_{Xii} - 2\sum_{j=1}^d D_{Yjj} Q_{ji}^2) \tag{48}$$

$$= \sum_{i=1}^d (D_{Xii}^2 + D_{Yii}^2) - 2 \sum_{i=1}^d \sum_{j=1}^d D_{Xii} D_{Yjj} Q_{ji}^2 \tag{49}$$

$$= \sum_{i=1}^d \sum_{j=1}^d Q_{ji}^2 (D_{Xii} - D_{Yjj})^2. \tag{50}$$

The last step makes use of $\sum_{i=1}^d Q_{ji}^2 = 1$ and $\sum_{j=1}^d Q_{ji}^2 = 1$. Similarly, using (42), we have,

$$\|A(X) - A(Y)\|_F^2 = \left\| \sqrt{D_X^2 + \frac{4}{\lambda} I} - Q^\top \sqrt{D_Y^2 + \frac{4}{\lambda} I} Q \right\|_F^2 \tag{51}$$

$$= \sum_{i=1}^d \sum_{j=1}^d Q_{ji}^2 (\sqrt{D_{Xii}^2 + \frac{4}{\lambda}} - \sqrt{D_{Yjj}^2 + \frac{4}{\lambda}})^2 \tag{52}$$

Assuming $\|X - Y\|_F > 0$ (otherwise (37) trivially holds), using (52) and (50), we have,

$$\frac{\|A(X) - A(Y)\|_F^2}{\|X - Y\|_F^2} = \frac{\sum_{i=1}^d \sum_{j=1}^d Q_{ji}^2 (\sqrt{D_{Xii}^2 + \frac{4}{\lambda}} - \sqrt{D_{Yjj}^2 + \frac{4}{\lambda}})^2}{\sum_{i=1}^d \sum_{j=1}^d Q_{ji}^2 (D_{Xii} - D_{Yjj})^2} \tag{53}$$

$$\leq \max_{i,j=1,\cdots,d,\, D_{Xii} \neq D_{Yjj}} \frac{(\sqrt{D_{Xii}^2 + \frac{4}{\lambda}} - \sqrt{D_{Yjj}^2 + \frac{4}{\lambda}})^2}{(D_{Xii} - D_{Yjj})^2} \tag{54}$$

Using lemma (2), we have,

$$\frac{\|A(X) - A(Y)\|_F^2}{\|X - Y\|_F^2} \le \max_{i,j=1,\cdots,d,\; D_{Xii} \neq D_{Yjj}} \frac{(\sqrt{D_{Xii}^2 + \frac{4}{\lambda}} - \sqrt{D_{Yjj}^2 + \frac{4}{\lambda}})^2}{(D_{Xii} - D_{Yjj})^2} \tag{55}$$

$$\le \max_{i,j=1,\cdots,d,\; D_{Xii} \neq D_{Yjj}} 1 - \frac{1}{\sqrt{(\frac{D_{Xii}^2}{\frac{4}{\lambda}} + 1)(\frac{D_{Yjj}^2}{\frac{4}{\lambda}} + 1)}} \tag{56}$$

$$\le 1 - \frac{1}{\sqrt{(\frac{\lambda}{4} \max_i D_{Xii}^2 + 1)(\frac{\lambda}{4} \max_j D_{Yjj}^2 + 1)}} \tag{57}$$

$$= 1 - (\frac{\lambda}{4} \Lambda_{max}(X)^2 + 1)^{-1/2} (\frac{\lambda}{4} \Lambda_{max}(Y)^2 + 1)^{-1/2}. \tag{58}$$

Therefore,

$$\frac{\|A(X) - A(Y)\|_F}{\|X - Y\|_F} \le \sqrt{1 - (\frac{\lambda}{4} \Lambda_{max}(X)^2 + 1)^{-1/2} (\frac{\lambda}{4} \Lambda_{max}(Y)^2 + 1)^{-1/2}} \tag{59}$$

$$\le 1 - \frac{1}{2} (\frac{\lambda}{4} \Lambda_{max}(X)^2 + 1)^{-1/2} (\frac{\lambda}{4} \Lambda_{max}(Y)^2 + 1)^{-1/2}. \tag{60}$$

$\square$

**Lemma 4.** *Under assumption (2), the output of the $k$-th and $(k+1)$-th AM step $Z_k^{AM}$, $Z_{k+1}^{AM}$ and $\Theta_{k+1}^{AM}$ satisfy the following inequality,*

$$\left\| Z_{k+1}^{AM} - \widehat{Z}_\lambda \right\|_F \le \left\| \Theta_{k+1}^{AM} - \widehat{\Theta}_\lambda \right\|_F \le C_\lambda \left\| Z_k^{AM} - \widehat{Z}_\lambda \right\|_F, \tag{61}$$

*where $0 < C_\lambda < 1$ is a constant depending on $\lambda$.*

*Proof.* The first part is easy to show, if we observe that in the second update step of AM (8), $\eta_{\rho/\lambda}$ is a contraction under metric $d(X,Y) = \|X - Y\|_F$. Therefore we have,

$$\left\| Z_{k+1}^{AM} - \widehat{Z}_\lambda \right\|_F = \left\| \eta_{\rho/\lambda}(\Theta_{k+1}^{AM}) - \eta_{\rho/\lambda}(\widehat{\Theta}_\lambda) \right\|_F \le \left\| \Theta_{k+1}^{AM} - \widehat{\Theta}_\lambda \right\|_F. \tag{62}$$

Next we will prove the second part. To simplify notation, we let $A(X) = \sqrt{X^\top X + \frac{4}{\lambda} I}$. Using the first update step of AM (7), we have,

$$\left\| \Theta_{k+1}^{AM} - \widehat{\Theta}_\lambda \right\|_F = \left\| \frac{1}{2} \left( -Y_{k+1} + \sqrt{Y_{k+1}^\top Y_{k+1} + \frac{4}{\lambda} I} \right) - \frac{1}{2} \left( -Y_\lambda + \sqrt{Y_\lambda^\top Y_\lambda + \frac{4}{\lambda} I} \right) \right\|_F \tag{63}$$

$$= \frac{1}{2} \left\| -(Y_{k+1} - Y_\lambda) + \left( \sqrt{Y_{k+1}^\top Y_{k+1} + \frac{4}{\lambda} I} - \sqrt{Y_\lambda^\top Y_\lambda + \frac{4}{\lambda} I} \right) \right\|_F \tag{64}$$

$$= \frac{1}{2} \left\| -(Y_{k+1} - Y_\lambda) + (A(Y_{k+1}) - A(Y_\lambda)) \right\|_F \tag{65}$$

$$\le \frac{1}{2} \|Y_{k+1} - Y_\lambda\|_F + \frac{1}{2} \|A(Y_{k+1}) - A(Y_\lambda)\|_F, \tag{66}$$

where $Y_{k+1} = \frac{1}{\lambda} \widehat{\Sigma} - Z_k^{AM}$ and $Y_\lambda = \frac{1}{\lambda} \widehat{\Sigma} - \widehat{Z}_\lambda$. The last derivation step makes use of the triangle inequality. Using lemma (3), we have,

$$\left\| \Theta_{k+1}^{AM} - \widehat{\Theta}_\lambda \right\|_F \le \frac{1}{2} \|Y_{k+1} - Y_\lambda\|_F + \frac{1}{2} \alpha_\lambda \|Y_{k+1} - Y_\lambda\|_F. \tag{67}$$

Therefore

$$\left\| \Theta_{k+1}^{AM} - \widehat{\Theta}_\lambda \right\|_F \le C_\lambda \|Y_{k+1} - Y_\lambda\|_F = C_\lambda \left\| Z_k^{AM} - \widehat{Z}_\lambda \right\|_F, \tag{68}$$

where

$$C_\lambda = \frac{1}{2} + \frac{1}{2} \alpha_\lambda = 1 - \frac{1}{4} (\frac{\lambda}{4} \Lambda_{max}(Y_{k+1})^2 + 1)^{-1/2} (\frac{\lambda}{4} \Lambda_{max}(Y_\lambda)^2 + 1)^{-1/2} \tag{69}$$

$$= 1 - (\lambda \Lambda_{max}(Y_{k+1})^2 + 4)^{-1/2} (\lambda \Lambda_{max}(Y_\lambda)^2 + 4)^{-1/2} \le 1, \tag{70}$$

$\Lambda_{max}(X)$ is the largest eigenvalue of $X$ in absolute value. The rest is to show that both $\Lambda_{max}(Y_\lambda)$ and $\Lambda_{max}(Y_{k+1})$ are bounded using assumption (2). For $\Lambda_{max}(Y_{k+1})$, we have,

$$\Lambda_{max}(Y_{k+1}) = \|Y_{k+1}\|_2 = \left\|\frac{1}{\lambda}\widehat{\Sigma} - Z_k^{AM}\right\|_2 \tag{71}$$

$$\leq \left\|\frac{1}{\lambda}\widehat{\Sigma}\right\|_2 + \|Z_k^{AM}\|_2 \tag{72}$$

$$\leq \frac{1}{\lambda}c_{\widehat{\Sigma}} + \left\|Z_k^{AM} - \widehat{Z}_\lambda\right\|_F + \left\|\widehat{Z}_\lambda\right\|_F. \tag{73}$$

Combining (62) and (68), we have,

$$\left\|Z_{k+1}^{AM} - \widehat{Z}_\lambda\right\|_F \leq \left\|\Theta_{k+1}^{AM} - \widehat{\Theta}_\lambda\right\|_F \leq C_\lambda \left\|Z_k^{AM} - \widehat{Z}_\lambda\right\|_F. \tag{74}$$

Therefore,

$$\left\|Z_k^{AM} - \widehat{Z}_\lambda\right\|_F \leq \left\|Z_{k-1}^{AM} - \widehat{Z}_\lambda\right\|_F \leq \cdots \leq \left\|Z_0^{AM} - \widehat{Z}_\lambda\right\|_F. \tag{75}$$

Continuing with (73), we have,

$$\Lambda_{max}(Y_{k+1}) \leq \frac{1}{\lambda}c_{\widehat{\Sigma}} + \left\|Z_k^{AM} - \widehat{Z}_\lambda\right\|_F + \left\|\widehat{Z}_\lambda\right\|_F \tag{76}$$

$$\leq \frac{1}{\lambda}c_{\widehat{\Sigma}} + \left\|Z_0^{AM} - \widehat{Z}_\lambda\right\|_F + \left\|\widehat{Z}_\lambda\right\|_F \tag{77}$$

$$\leq \frac{1}{\lambda}c_{\widehat{\Sigma}} + \|Z_0^{AM}\|_F + 2\left\|\widehat{Z}_\lambda\right\|_F. \tag{78}$$

Since $\widehat{Z}_\lambda$ is the minimizer of a strongly convex function, its norm is bounded. And we also have $\|Z_0^{AM}\|_F$ bounded in Algorithm (1), so $\Lambda_{max}(Y_{k+1})$ is bounded above whenever $\lambda < \infty$. For $\Lambda_{max}(Y_\lambda)$, we have,

$$\Lambda_{max}(Y_\lambda) = \left\|\frac{1}{\lambda}\widehat{\Sigma} - \widehat{Z}_\lambda\right\|_2 \tag{79}$$

$$\leq \left\|\frac{1}{\lambda}\widehat{\Sigma}\right\|_2 + \left\|\widehat{Z}_\lambda\right\|_2 \tag{80}$$

$$\leq \frac{1}{\lambda}c_{\widehat{\Sigma}} + \left\|\widehat{Z}_\lambda\right\|_2. \tag{81}$$

Therefore both $\Lambda_{max}(Y_\lambda)$ and $\Lambda_{max}(Y_{k+1})$ are bounded in (70), *i.e.* $0 < C_\lambda < 1$ is a constant only depending on $\lambda$.

$$\square$$

**Theorem 1.** *Under the assumptions 1 & 2, if $\rho \asymp \sqrt{\frac{\log d}{m}}$, where $\rho$ is the $l_1$ penalty, $d$ is the dimension of problem and $m$ is the number of samples, the Alternate Minimization algorithm has linear convergence rate for optimization objective defined in (6). The $k^{th}$ iteration of the AM algorithm satisfies,*

$$\left\|\Theta_k^{AM} - \Theta^*\right\|_F \leqslant C_\lambda \left\|\Theta_{k-1}^{AM} - \widehat{\Theta}_\lambda\right\|_F + \mathcal{O}_\mathbb{P}\left(\sqrt{\frac{(\log d)/m}{\min(\frac{1}{(d+s)}, \frac{\lambda}{d^2})}}\right), \tag{11}$$

*where $0 < C_\lambda < 1$ is a constant depending on $\lambda$.*

*Proof.* *(1) Error between $\widehat{\Theta}_\lambda$ and $\widehat{\Theta}_\mathcal{G}$*

Combining the following two equations:

$$f(\widehat{\Theta}_\lambda, \widehat{Z}_\lambda; \rho, \lambda) = \min_{\Theta, Z} f(\Theta, Z; \rho, \lambda) \leqslant \min_\Theta f(\Theta, Z = \Theta; \rho, \lambda) = \min_\Theta \mathcal{G}(\Theta; \rho) = \mathcal{G}(\widehat{\Theta}_\mathcal{G}; \rho),$$

$$f(\Theta, Z; \rho, \lambda) = \mathcal{G}(\Theta; \rho) + \rho(\|Z\|_1 - \|\Theta\|_1) + \frac{1}{2}\lambda\|Z - \Theta\|_F^2,$$

we have:

$$0 \leqslant \mathcal{G}(\widehat{\Theta}_\lambda; \rho) - \mathcal{G}(\widehat{\Theta}_\mathcal{G}; \rho) \leqslant \rho(\left\|\widehat{\Theta}_\lambda\right\|_1 - \left\|\widehat{Z}_\lambda\right\|_1).$$

Note that by the optimality condition, $\nabla_z f(\widehat{\Theta}_\lambda, \widehat{Z}_\lambda, \rho, \lambda) = 0$, we have the fixed point equation
$$\widehat{Z}_\lambda = \eta_{\rho/\lambda}(\widehat{\Theta}_\lambda).$$
Therefore, $\left\|\widehat{\Theta}_\lambda\right\|_1 - \left\|\widehat{Z}_\lambda\right\|_1 \leqslant \frac{\rho d^2}{\lambda}$ and we have:
$$0 \leqslant \mathcal{G}(\widehat{\Theta}_\lambda; \rho) - \mathcal{G}(\widehat{\Theta}_{\mathcal{G}}; \rho) \leqslant \frac{\rho^2 d^2}{\lambda}. \tag{82}$$

Since $\mathcal{G}$ is $\sigma_{\mathcal{G}}$-strongly convex, where $\sigma_{\mathcal{G}}$ is independent of the sample covariance matrix $\widehat{\Sigma}^*$ as the hessian of $\mathcal{G}$ is independent of $\widehat{\Sigma}^*$.
$$\frac{\sigma_{\mathcal{G}}}{2}\left\|\widehat{\Theta}_\lambda - \widehat{\Theta}_{\mathcal{G}}\right\|_F^2 + \langle \nabla\mathcal{G}(\widehat{\Theta}_{\mathcal{G}}; \rho), \widehat{\Theta}_\lambda - \widehat{\Theta}_{\mathcal{G}} \rangle \leqslant \mathcal{G}(\widehat{\Theta}_\lambda; \rho) - \mathcal{G}(\widehat{\Theta}_{\mathcal{G}}; \rho). \tag{83}$$
Therefore,
$$\left\|\widehat{\Theta}_\lambda - \widehat{\Theta}_{\mathcal{G}}\right\|_F \leqslant \sqrt{\frac{2\rho^2 d^2}{\lambda \sigma_{\mathcal{G}}}} = \rho d \sqrt{\frac{2}{\lambda \sigma_{\mathcal{G}}}} = O\left(\rho d \sqrt{\frac{1}{\lambda}}\right) \tag{84}$$

$\square$

*Proof. (2) Error between $\widehat{\Theta}_{\mathcal{G}}$ and $\Theta^*$*

**Corollary 5** (Theorem 1. of Rothman et al. (2008)). *Let $\widehat{\Theta}_{\mathcal{G}}$ be the minimizer for the optimization objective $\mathcal{G}(\Theta; \rho)$. Under Assumptions 1 & 2, if $\rho \asymp \sqrt{\frac{\log d}{m}}$,*
$$\left\|\widehat{\Theta}_{\mathcal{G}} - \Theta^*\right\|_F = \mathcal{O}_{\mathbb{P}}\left(\sqrt{\frac{(d+s)\log d}{m}}\right) \tag{85}$$

$\square$

*(3) Error between $\Theta_k^{AM}$ and $\Theta^*$*

Under the conditions in Corollary 5, we use triangle inequality to combine the above results with Corollary 5 and Lemma 4.
$$\left\|\Theta_k^{AM} - \Theta^*\right\|_F \leqslant \left\|\Theta_k^{AM} - \widehat{\Theta}_\lambda\right\|_F + \left\|\widehat{\Theta}_\lambda - \widehat{\Theta}_{\mathcal{G}}\right\|_F + \left\|\widehat{\Theta}_{\mathcal{G}} - \Theta^*\right\|_F \tag{86}$$
$$\leqslant C_\lambda \left\|\Theta_{k-1}^{AM} - \widehat{\Theta}_\lambda\right\|_F + O\left(\rho d \sqrt{\frac{1}{\lambda}}\right) + \mathcal{O}_{\mathbb{P}}\left(\sqrt{\frac{(d+s)\log d}{m}}\right) \tag{87}$$
$$\leqslant C_\lambda \left\|\Theta_{k-1}^{AM} - \widehat{\Theta}_\lambda\right\|_F + \mathcal{O}_{\mathbb{P}}\left(\sqrt{\frac{(d+s)\log d}{m.\min(1, \frac{(d+s)\lambda}{d^2})}}\right) \tag{88}$$
$$\leqslant C_\lambda \left\|\Theta_{k-1}^{AM} - \widehat{\Theta}_\lambda\right\|_F + \mathcal{O}_{\mathbb{P}}\left(\sqrt{\frac{(\log d)/m}{\min(\frac{1}{(d+s)}, \frac{\lambda}{d^2})}}\right) \tag{89}$$

## C  EXPERIMENTAL DETAILS

This section contains the detailed settings used in the experimental evaluation section.

### C.1  SYNTHETIC DATASET GENERATION

For sections 5.1 and 5.2, the synthetic data was generated based on the procedure described in Rolfs et al. (2012). A $d$ dimensional precision matrix $\Theta$ was generated by initializing a $d \times d$ matrix with its off-diagonal entries sampled i.i.d. from a uniform distribution $\Theta_{ij} \sim \mathcal{U}(-1, 1)$. These entries were then set to zero based on the sparsity pattern of the corresponding Erdos-Renyi random graph with a certain probability $p$. Finally, an appropriate multiple of the identity matrix was added to the current matrix, so that the resulting matrix had the smallest eigenvalue as 1. In this way, $\Theta$ was ensured to be

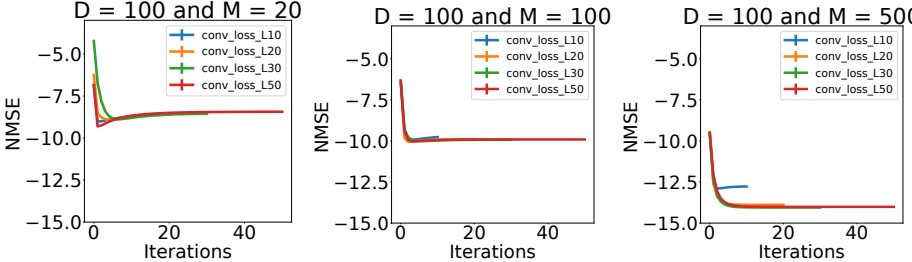

Figure 8: Varying the number of unrolled iterations. The results are averaged over $1000$ test graphs. The $L$ variable is the number of unrolled iterations. We observe that the higher number of unrolled iterations better is the performance.

a well-conditioned, sparse and positive definite matrix. This matrix was then used in the multivariate Gaussian distribution $\mathcal{N}(0, \Theta^{-1})$, to obtain $M$ i.i.d samples.

## C.2 EXPERIMENT DETAILS: BENEFIT OF DATA-DRIVEN GRADIENT-BASED ALGORITHM

Figure(2): The plots are for the ADMM method on the Erdos-Renyi graphs (fixed sparsity $p = 0.1$) with dimension $D = 100$ and number of samples $M = 100$. The results are averaged over $100$ test graphs with 10 sample batches per graph. The std-err = $\sigma / \sqrt{1000}$ is shown. Refer appendix(C.1) for more details on data generation process.

## C.3 EXPERIMENT DETAILS: EXPENSIVE HYPERPARAMETERS TUNING

Table(1) shows the final NMSE values for the ADMM method on the random graph (fixed sparsity $p = 0.1$) with dimension $D = 100$ and number of samples $M = 100$. We fixed the initialization parameter of $\Theta_0$ as $t = 0.1$ and chose appropriate update rate $\alpha$ for $\lambda$. It is important to note that the NMSE values are very sensitive to the choice of $t$ as well. These parameter values changed substantially for a new problem setting. Refer appendix(C.1) for more details on data generation process.

## C.4 EXPERIMENT DETAILS: CONVERGENCE ON SYNTHETIC DATASETS

Figure(4) experiment details: Figure(4) shows the NMSE comparison plots for fixed sparsity and mixed sparsity synthetic Erdos-renyi graphs. The dimension was fixed to $D = 100$ and the number of samples vary as $M = [20, 100, 500]$. The top row has the sparsity probability $p = 0.5$ for the Erdos-Renyi random graph, whereas for the bottom row plots, the sparsity probabilities are uniformly sampled from $\sim \mathcal{U}(0.05, 0.15)$. For finetuning the traditional algorithms, a validation dataset of 10 graphs was used. For the GLAD algorithm, 10 training graphs were randomly chosen and the same validation set was used.

## C.5 GLAD: ARCHITECTURE DETAILS FOR SECTION(5.2)

GLAD parameter settings: $\rho_{nn}$ was a 4 layer neural network and $\Lambda_{nn}$ was a 2 layer neural network. Both used 3 hidden units in each layer. The non-linearity used for hidden layers was $\tanh$, while the final layer had sigmoid ($\sigma$) as the non-linearity for both, $\rho_{nn}$ and $\Lambda_{nn}$ (refer Figure 3). The learnable offset parameter of initial $\Theta_0$ was set to $t = 1$. It was unrolled for $L = 30$ iterations. The learning rates were chosen to be around $[0.01, 0.1]$ and multi-step LR scheduler was used. The optimizer used was 'adam'. The best nmse model was selected based on the validation data performance. Figure(8) explores the performance of GLAD on using varying number of unrolled iterations $L$.

## C.6 ADDITIONAL NOTE OF HYPER-PARAMETER FINETUNING FOR TRADITIONAL METHODS

Figure(1) shows the average NMSE values over 100 test graphs obtained by the ADMM algorithm on the synthetic data for dimension $D = 100$ and $M = 100$ samples as we vary the values of penalty parameter $\rho$ and lagrangian parameter $\lambda$. The offset parameter for $\Theta_0$ was set to $t = 0.1$. The NMSE

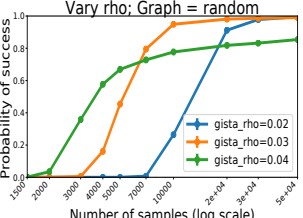 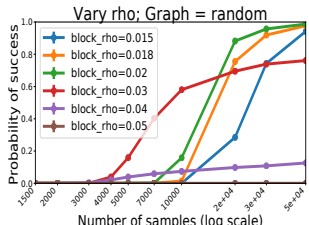 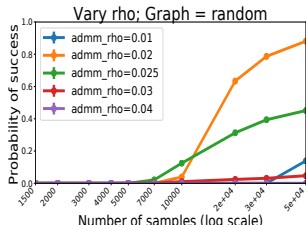

Figure 9: We attempt to illustrate how the traditional methods are very sensitive to the hyperparameters and it is a tedious exercise to finetune them. The problem setting is same as described in section(5.3). For all the 3 methods shown above, we have already tuned the algorithm specific parameters to a reasonable setting. Now, we vary the $L_1$ penalty term $\rho$ and can observe that how sensitive the probability of success is with even slight change of $\rho$ values.

values are very sensitive to the choice of $t$ as well. These parameter values changes substantially for a new problem setting. G-ISTA and BCD follow similar trends.

Additional plots highlighting the hyperparameter sensitivity of the traditional methods for model selection consistency experiments. Refer figure(9).

### C.7 TOLERANCE OF NOISE: EXPERIMENT DETAILS

Details for experiments in figure(5). Two different graph types were chosen for this experiment which were inspired from Ravikumar et al. (2011). In the 'grid' graph setting, the edge weight for different precision matrices were uniformly sampled from $w \sim \mathcal{U}(0.12, 0.25)$. The edges within a graph carried equal weights. The other setting was more general, where the graph was a random Erdos-Renyi graph with probability of an edge was $p = 0.05$. The off-diagonal entries of the precision matrix were sampled uniformly from $\sim \mathcal{U}[0.1, 0.4]$. The parameter settings for GLAD were the same as described in Appendix C.5. The model with the best PS performance on the validation dataset was selected. train/valid/test=10/10/100 graphs were used with 10 sample batches per graph.

### C.8 GLAD: COMPARISON WITH OTHER DEEP LEARNING BASED METHODS

Table(3) shows AUC (with std-err) comparisons with the DeepGraph model. For experiment settings, refer Table 1 of Belilovsky et al. (2017). Gaussian Random graphs with sparsity $p = 0.05$ were chosen and edge values sampled from $\sim \mathcal{U}(-1, 1)$. GLAD was trained on only 10 graphs with 5 sample batches per graph. The dimension of the problem is $D = 39$. The architecture parameter choices of GLAD were the same as described in Appendix C.5 and it performs consistently better along all the settings by a significant AUC margin.

### C.9 SYNTREN GENE EXPRESSION SIMULATOR DETAILS

The SynTReN Van den Bulcke et al. (2006) is a synthetic gene expression data generator specifically designed for analyzing the structure learning algorithms. The topological characteristics of the synthetically generated networks closely resemble the characteristics of real transcriptional networks. The generator models different types of biological interactions and produces biologically plausible synthetic gene expression data enabling the development of data-driven approaches to recover the underlying network.

The SynTReN simulator details for section(5.5). For performance evaluation, a connected Erdos-Renyi graph was generated with probability as $p = 0.05$. The precision matrix entries were sampled from $\Theta_{ij} \sim \mathcal{U}(0.1, 0.2)$ and the minimum eigenvalue was adjusted to $1$ by adding an appropriate multiple of identity matrix. The SynTReN simulator then generated samples from these graphs by incorporating biological noises, correlation noises and other input noises. All these noise levels were sampled uniformly from $\sim \mathcal{U}(0.01, 0.1)$. The figure(6) shows the NMSE comparisons for a fixed dimension $D = 25$ and varying number of samples $M = [10, 25, 100]$. The number of training/validation graphs were set to 20/20 and the results are reported on 100 test graphs. In these experiments, only 1 batch of $M$ samples were taken per graph to better mimic the real world setting.

Figure(7) visualizes the edge-recovery performance of the above trained `GLAD` models on a subnetwork of true Ecoli bacteria data.which contains 30 edges and $D = 43$ nodes. The Ecoli subnetwork graph was fed to the SynTReN simulator and $M$ samples were obtained. SynTReN's noise levels were set to 0.05 and the precision matrix edge values were set to $w = 0.15$. For the `GLAD` models, the training was done on the same settings as the gene-data NMSE plots with $D = 25$ and on corresponding number of samples $M$.

## C.10 COMPARISON WITH ADMM OPTIMIZATION BASED UNROLLED ALGORITHM

In order to find the best unrolled architecture for sparse graph recovery, we considered many different optimization techniques and came up with their equivalent unrolled neural network based deep model. In this section, we compare with the closest unrolled deep model based on ADMM optimization, (`ADMMu`), and analyze how it compares to `GLAD`. Appendix C.11 lists down further such techniques for future exploration.

*Unrolled model for ADMM:* Algorithm 2 describes the unrolled model `ADMMu` updates. $\rho_{nn}$ was a 4 layer neural network and $\Lambda_{nn}$ was a 2 layer neural network. Both used 3 hidden units in each layer. The non-linearity used for hidden layers was tanh, while the final layer had sigmoid ($\sigma$) as the non-linearity for both ,$\rho_{nn}$ and $\Lambda_{nn}$. The learnable offset parameter of initial $\Theta_0$ was set to $t = 1$. It was unrolled for $L = 30$ iterations. The learning rates were chosen to be around $[0.01, 0.1]$ and multi-step LR scheduler was used. The optimizer used was 'adam'.

Figure 10 compares `GLAD` with `ADMMu` on the convergence performance with respect to synthetically generated data. The settings were kept same as described in Figure 4. As evident from the plots, we see that `GLAD` consistently performs better than `ADMMu`. We had similar observations for other set of experiments as well. Hence, we chose AM based unrolled algorithm over ADMM's as it works better empirically and has less parameters.

Although, we are not entirely confident but we hypothesize the reason for above observations as follows. In the ADMM update equations (4 & 5), both the Lagrangian term and the penalty term are intuitively working together as a 'function' to update the entries $\Theta_{ij}, Z_{ij}$. Observe that $U_k$ can be absorbed into $Z_k$ and/or $\Theta_k$ and we expect our neural networks to capture this relation. We thus expect `GLAD` to work at least as good as `ADMMu`. In our formulation of unrolled `ADMMu` (Algorithm 2) the update step of $U$ is not controlled by neural networks (as the number of parameters needed will be substantially larger) which might be the reason of it not performing as well as `GLAD`. Our empirical evaluations

---

**Algorithm 2:** `ADMMu`

**Function** `ADMMu-cell`$(\widehat{\Sigma}, \Theta, Z, U, \lambda)$ **:**

$\quad \lambda \leftarrow \Lambda_{nn}(\|Z - \Theta\|_F^2, \lambda)$

$\quad Y \leftarrow \lambda^{-1}\widehat{\Sigma} - Z + U$

$\quad \Theta \leftarrow \frac{1}{2}\left(-Y + \sqrt{Y^\top Y + \frac{4}{\lambda}I}\right)$

$\quad$ **For** *all* $i, j$ **do**

$\quad\quad \rho_{ij} = \rho_{nn}(\Theta_{ij}, \widehat{\Sigma}_{ij}, Z_{ij}, \lambda)$

$\quad\quad Z_{ij} \leftarrow \eta_{\rho_{ij}}(\Theta_{ij} + U_{ij})$

$\quad U \leftarrow U + \Theta - Z$

$\quad$ **return** $\Theta, Z, U, \lambda$

**Function** `ADMMu`$(\widehat{\Sigma})$ **:**

$\quad \Theta_0 \leftarrow (\widehat{\Sigma} + tI)^{-1}, \lambda_0 \leftarrow 1$

$\quad$ **For** $k = 0$ *to* $K - 1$ **do**

$\quad\quad \Theta_{k+1}, Z_{k+1}, U_{k+1}, \lambda_{k+1}$

$\quad\quad \leftarrow$ `ADMMu-cell`$(\widehat{\Sigma}, \Theta_k$

$\quad\quad\quad\quad , Z_k, U_k, \lambda_k)$

$\quad$ **return** $\Theta_K, Z_K$

---

corroborate this logic that just by using the penalty term we can maintain all the desired properties and learn the problem dependent 'functions' with a small neural network.

## C.11 DIFFERENT DESIGNS TRIED FOR DATA-DRIVEN ALGORITHM

We tried multiple unrolled parameterizations of the optimization techniques used for solving the graphical lasso problem which worked to varying levels of success. We list here a few, in interest for helping researchers to further pursue this recent and novel approach of data-driven algorithm designing.

1. ADMM + ALISTA parameterization: The threshold update for $Z_{k+1}^{AM}$ can be replaced by ALISTA network Liu et al. (2018). The stage I of ALISTA is determining W, which is trivial in our case as $D = I$. So, we get $W = I$. Thus, combining ALISTA updates along with AM's we get an interesting unrolled algorithm for our optimization problem.

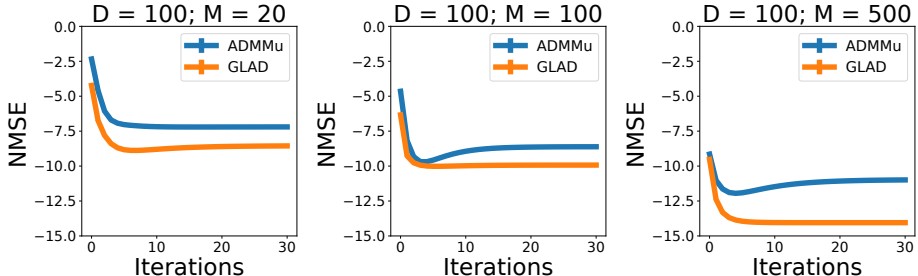

Figure 10: Convergence on Erdos-random graphs with fixed sparsity ($p = 0.1$). All the settings are same as the fixed sparsity case described in Figure 4. We see that the AM based parameterization 'GLAD' consistently performs better than the ADMM based unrolled architecture 'ADMMu'.

2. G-ISTA parameterization: We parameterized the line search hyperparameter $c$ as well as replaced the next step size determination step by a problem dependent neural network of Algorithm(1) in Rolfs et al. (2012). The main challenge with this parameterization is to main the PSD property of the intermediate matrices obtained. Learning appropriate parameterization of line search hyperparameter such that PSD condition is maintained remains an interesting aspect to investigate.
3. Mirror Descent Net: We get a similar set of update equations for the graphical lasso optimization. We identify some learnable parameters, use neural networks to make them problem dependent and train them end-to-end.
4. For all these methods we also tried unrolling the neural network as well. In our experience we found that the performance does not improve much but the convergence becomes unstable.

### C.12 RESULTS ON REAL DATA

We use the real data from the 'DREAM 5 Network Inference challenge' (Marbach et al., 2012). This dataset contains 3 compendia that were obtained from microorganisms, some of which are pathogens of clinical relevance. Each compendium consists of hundreds of microarray experiments, which include a wide range of genetic, drug, and environmental perturbations. We test our method for recovering the true E.coli network from the gene expression values recorded by doing actual microarray experiments.

The E.coli dataset contains $4511$ genes and $805$ associated microarray experiments. The true underlying network has $2066$ discovered edges and $150214$ pairs of nodes do not have an edge between them. There is no data about the remaining edges. For our experiments, we only consider the discovered edges as the ground truth, following the challenge data settings. We remove the genes that have zero degree and then we get a subset of $1081$ genes. For our predictions, we ignore the direction of the edges and only consider retrieving the connections between genes.

We train the GLAD model using the SynTReN simulator on the similar settings as described in Appendix C.9. Briefly, GLAD model was trained on D=50 node graphs sampled from Erdos-Renyi graph with sparsity probability $\sim U(0.01, 0.1)$, noise levels of SynTReN simulator sampled from $\sim U(0.01, 0.1)$ and $\Theta_{ij} \sim U(0.1, 0.2)$). The model was unrolled for 15 iterations. This experiment also evaluates GLAD's ability to generalize to different distribution from training as well as scaling ability to more number of nodes.

We report the AUC scores for E.coli network in Table 4 . We can see that GLAD improves over the other competing methods in terms of Area Under the ROC curve (AUC). We understand that it is challenging to model real datasets due to the presence of many unknown latent extrinsic factors, but we do observe an advantage of using data-driven parameterized algorithm approaches.

### C.13 SCALING FOR LARGE MATRICES

| Methods | BCD | GISTA | **GLAD** |
|---------|-----|-------|----------|
| AUC | 0.548 | 0.541 | **0.572** |

Table 4: GLAD vs other methods for the DREAM network inference challenge real E.Coli data.

We have shown in our experiments that we can train GLAD on smaller number of nodes and get reasonable results for recovering graph structure with considerably larger nodes (AppendixC.12). Thus, in this section, we focus on scaling up on the inference/test part.

With the current GPU implementation, we can can handle around 10,000 nodes for inference. For problem sizes with more than 100,000 nodes, we propose to use the randomized algorithm techniques given in Kannan & Vempala (2017). Kindly note that scaling up GLAD is our ongoing work and we just present here one of the directions that we are exploring. The approach presented below is to give some rough idea and may contain loose ends.

Randomized algorithms techniques are explained elaborately in Kannan & Vempala (2017). Specifically, we will use some of their key results

- P1. (Theorem 2.1) We will use the length-squared sampling technique to come up with low-rank approximations
- P2. (Theorem 2.5) For any large matrix $A \in R^{m \times n}$, we can use approximate it as $A \approx CUR$, where $C \in R^{m \times r}, U \in R^{s \times r}, R \in R^{r \times m}$.
- P3. (Section 2.3) For any large matrix $A \in R^{m \times n}$, we can get its approximate SVD by using the property $E(R^T R) = A^T A$ where $R$ is a matrix obtained by length-squared sampling of the rows of matrix $A$.

The steps for doing approximate AM updates, i.e. of equations(7, 8). Using property P3, we can approximate $Y^T Y \approx R^T R$.

$$\sqrt{Y^\top Y + \tfrac{4}{\lambda} I} \approx \sqrt{R^T R + \tfrac{4}{\lambda} I} \approx V \sqrt{\left(\Sigma^2 + \tfrac{4}{\lambda}\right)} V^T \tag{90}$$

where $V$ is the right singular vectors of R. Thus, we can combine this approximation with the sketch matrix approximation of $Y \approx CUR$ to calculate the update in equation(7). Equation(8) is just a thresholding operation and can be done efficiently with careful implementation. We are looking in to the experimental as well as theoretical aspects of this approach.

We are also exploring an efficient distributed algorithm for GLAD. We are investigating into parallel MPI based algorithms for this task (https://stanford.edu/~boyd/admm.html is a good reference point). We leverage the fact that the size of learned neural networks are very small, so that we can duplicate them over all the processors. This is also an interesting future research direction.

